



# Combined impacts of climate change and human activities on blue and green water resources in the high-intensity development watershed

Xuejin Tan [1], Bingjun Liu [2*], Xuezhi Tan [2, 3*], Zeqin Huang[2], Jianyu Fu[2]

[1] School of Geography and Planning, Sun Yat-sen University, Guangzhou, 510006, PR China

[2] Center of Water Resources and Environment, School of Civil Engineering, Sun Yat-sen University, Guangzhou, 510275, PR China

[3] Southern Marine Science and Engineering Guangdong Laboratory (Zhuhai), Sun Yat-sen University, Zhuhai, 519082, PR China

* Corresponding authors: Bingjun Liu (liubj@mail.sysu.edu.cn)

Xuezhi Tan (tanxuezhi@mail.sysu.edu.cn)



**Abstract**

2       Sustainable management of blue and green water resources is vital for the stability and

sustainability of watershed ecosystems. Although there has been extensive attention to blue water
($BW$) which is closely related to human beings, the relevance of green water ($GW$) for ecosystem
security is typically disregarded in water resource evaluations. Specifically, there is a scarcity of
comprehensive study on the detection and attribution of variation of blue and green water in the
Dongjiang River Basin (DRB), an important source for regional water supply in the Guangdong-
Hong Kong-Macao Greater Bay Area (GBA) of China. Here we assess the variations of $BW$ and
$GW$ scarcity, quantify the impacts of climate change and land use change on $BW$ and $GW$ in DRB
using a multi-water-flux calibrated Soil and Water Assessment Tool (SWAT). Results show that
$BW$ and green water storage ($GWS$) in DRB increased slowly with a rate of 0.14 and 0.015 mm a$^{-1}$
, respectively, while green water flow ($GWF$) decreased significantly at a rate of -0.21 mm a$^{-1}$.
The degree of $BW$ and $GW$ scarcity in DRB is low, and the per capita water resources in more than
80% of DRB exceed 1700 m$^3$ capita$^{-1}$ a$^{-1}$. Attribution results show that 88.0%, 88.5%, and 39.4%
of changes in $BW$, $GWF$, and $GW$S results from climate change, respectively. Both climate change
and land use change have decrease $BW$, while climate change (land use change) decrease (increase)
$GWF$ in DRB. These findings can guide to optimize the allocation of blue and green water
resources between upper reach and lower reach areas in DRB and further improve the
understanding of blue and green water evolution patterns in humid regions.
**Key words:** Blue and green water; Water scarcity; Climate change, Land use change; Water flow;
Dongjiang River Basin
**1 Introduction**

23       Land use change (LUCC) and climate variability may alter hydrological processes in



watersheds (Chagas et al., 2022; Konapala et al., 2020; Xuezhi Tan et al., 2022), which
successively affect variations of regional water resources (Hoek van Dijke et al., 2022; Pokhrel et
al., 2021; Stocker et al., 2023), potentially leading to ecosystem degradation and severe water
shortage crises (Aghakhani Afshar et al., 2018; Zuo et al., 2015). With the development of society
and the economy, there is an increasing need for more water to accommodation human needs,
encompassing agricultural, domestic, and industrial water usage. Water scarcity and
spatiotemporal mismatch between regional water supply and demand in certain regions are
becoming increasingly severe, significantly affecting the sustainable development in these regions
(Cook et al., 2014). Quantifying water resources under a changing environment is crucial for
guiding efficient and sustainable water use.

Previous studies on water resource assessment have explored the effects of climate change

and anthropogenic on available water resources, including streamflow (Tan and Gan, 2015; Tan et
al., 2023; Xin et al., 2019), baseflow (Ficklin et al., 2016; Tan et al., 2020), lake water (Acero
Triana and Ajami, 2022; Tao et al., 2020), and groundwater (Han et al., 2020). Falkenmark and
Rockström (2006) introduce a novel perspective on water resource assessment by categorizing
water resources into $BW$ and $GW$. $BW$ is the total of deep aquifers recharge and river streamflow,
such as water in lakes, and rivers. Water users such as industries, agriculture, and municipal users
can directly utilize $BW$. On the contrary, $GW$ is the portion of precipitation that is not streamflow
and is temporarily retained in the soil before eventually being released back into the air by



evapotranspiration. *GW* encompasses both green water flow (*GWF*) and green water storage (*GWS*)
(Veettil and Mishra, 2018; Zang and Liu, 2013). Traditional water resource assessments
concentrating on available water resources. Only consider *BW* but neglect *GW* (Dai et al., 2022),
although *GW* is also essential. *GW* supplies about 80% of total water resources, sustaining crops
growth and the sustainable development of forest and grasslands ecosystems in arid regions or
during dry seasons (Li et al., 2018; Schuol et al., 2008). The green water scarcity can lead to
ecosystem degradation and intensify competition between human needs and ecosystems for water
resources (Falkenmark et al., 2003; Veettil and Mishra, 2018). Compared to traditional streamflow
assessment methods, water resource scarcity assessment methods based on the framework of *BW*
and *GW* are more appropriate for maintaining sustainable water resource management (Cooper et
al., 2022; Liu et al., 2017). Recently, some researches have characterized water scarcity by
assessing variations of *BW* and *GW*. Veettil and Mishra (2020) assess blue water scarcity and green
water scarcity to show the water security status of counties in the United States. Hoekstra et al.
(2012) uses the concept of *BW* footprint to study water scarcity issues. Schyns et al. (2019) uses
the conception of *GW* footprint to investigate green water scarcity and found that the increasingly
severe shortage of *GW* poses a significant threat to natural ecosystems.
Impacts of climate change and anthropogenic on the hydrological cycle processes in
watersheds have attracted widespread attention (Chouchane et al., 2020; Cooper et al., 2022;
Sherwood and Fu, 2014; Tan and Gan, 2015; Xuejin Tan et al., 2022; Veettil and Mishra, 2016).



Changes in land use alter the underlying surface conditions. For example, afforestation or
deforestation may exacerbate or alleviate global or regional climate change, and thus affect
hydrological cycle processes (Bai et al., 2020; Lian et al., 2020; Qiu et al., 2023). Changes in land
use often lead to alterations in land-atmosphere interactions, and vegetation cover changes are
essential for regulating climate systems and land ecosystems (Foley et al., 2005; Huang et al.,
2020). Large-scale greening could modify geophysical interactions between the atmosphere and
the ground, impacting larger or local regional hydrological cycles. Land degradation (Walters and
Babbar-Sebens, 2016), deforestation (Lee et al., 2011), and urbanization (Mohan and Kandya,
2015; Zhang et al., 2018) also have far-reaching effects on climate and hydrological cycle.
Climate change is also crucial to the variations in $BW$ and $GW$ resources. Precipitation is the
source of $BW$ and $GW$, and factors such as temperature, solar radiation, and potential
evapotranspiration significantly influence the changes of $BW$ and $GW$ in the basins, especially in
$GWF$ (Pandey et al., 2019; Schewe et al., 2014). For a single watershed, $BW$ depends directly on
precipitation and evapotranspiration ($GWF$) (Shen et al., 2017; Vano et al., 2012). Furthermore,
precipitation intensity can have a significant impact on the redistribution of precipitation and $BW$,
as well as $GW$, by altering infiltration and runoff generation processes (Eekhout et al., 2018;
Nearing et al., 2005). Therefore, it is crucial to quantify the effects of climate change and LUCC
on $BW$ and $GW$ resources in a basin for effective water resource planning and management.
The Dongjiang River Basin (DRB) is a crucial water source for core cities in GBA, such as



Shenzhen, Hong Kong, and Huizhou. Given the significant *BW* demand from agriculture, urban
areas, and industry, as well as the *GW* demand from over 18,000 km$^2$ of forested land, the water
resource stress in DRB is extremely high, although DRB is located in the wet South China (Liu et
al., 2018). The growing mismatch between increasing water demand and decreasing water supply,
along with seasonal and pollution-induced water scarcity issues, is becoming increasingly
prominent (Yang et al., 2018). Currently, the majority of studies on water resources of DRB focus
on changes and scarcity of surface water and groundwater (*BW*) while overlook the critical role
and spatiotemporal variations of *GW* (Huang et al., 2022; Jiang et al., 2023; Jiefeng Wu et al.,
2021). With the high-intensity urbanization and climate change in DRB, changes of *BW* and *GW*
resource in DRB remain unknowns.
This research aims to analyze the influence of climate change and LUCC on *BW* and *GW* in
DRB. The objectives of this research are (a) to build the SWAT model for DRB, (b) to
quantitatively evaluate the spatial and temporal variation of *BW* and *GW* in DRB, (c) to assess the
status of water scarcity in DRB using the framework of *BW* and *GW* resources, (d) to estimate the
effects of climate change and LUCC on *BW* and *GW* in DRB.
**2 Materials and methods**
2.1 Study area
The Dongjiang River serves as a important tributary of the Pearl River, positioned between



longitude 113°25'-115°52'E and latitude 22°26'-25°12'N. It originates in Xunwu County, Jiangxi
Province, flows through Jiangxi and Guangdong provinces, and goes across major cities including
Longchuan, Heyuan, Dongguan, and Shenzhen. The main stem of the Dongjiang River spans a
total length of 562 km, covering a watershed area of $3.5×104$ km$^2$. DRB is situated within the
subtropical monsoon climate zone with adequate precipitation and high temperatures. The average
annual precipitation ranges from 1500-2400 mm, and the average temperature of the basin is
approximately 21°C (Jiefeng Wu et al., 2019). The altitude of the basin decreases from northeast
to southwest. The upper reaches of DRB are dominated by mountains and hills, the middle reaches
of DRB are dominated by hills and plains, and the lower reaches of DRB are dominated by plains.
Previous hydrological simulation studies of DRB mainly use the Boluo hydrometric station
as the outlet of the watershed (He et al., 2013; Jiefeng Wu et al., 2019), so this research only
analyze the area of DRB where water flows to the Boluo station (Fig. 1). The Boluo hydrometric
station is the main control station in the lower reaches of the Dongjiang. The Boluo hydrometric
station occupy a drainage area of 25,325 km$^2$, which is 71.7% of the overall DRB. Since the 1950s,
more than 896 reservoirs, ponds, dams, and other water conservancy facilities have been
constructed in DRB. Among them, the Baipenzhu Reservoir, Fengshuiba Reservoir, and
Xinfengjiang Reservoir are the main reservoirs in the basin with a cumulative storage capacity of
approximately 17,048 million m$^3$. The Dongjiang-Shenzhen Water Supply Project constructed in
1964 diverts water from the Dongjiang River to Shenzhen and Hong Kong for providing fresh
water resources for municipal use. Over 70% of Hong Kong's freshwater supply comes from the
Dongjiang River. Therefore, it is crucial to comprehend the shifts in water resources within DRB
for projecting future available water resources for the development of GBA.

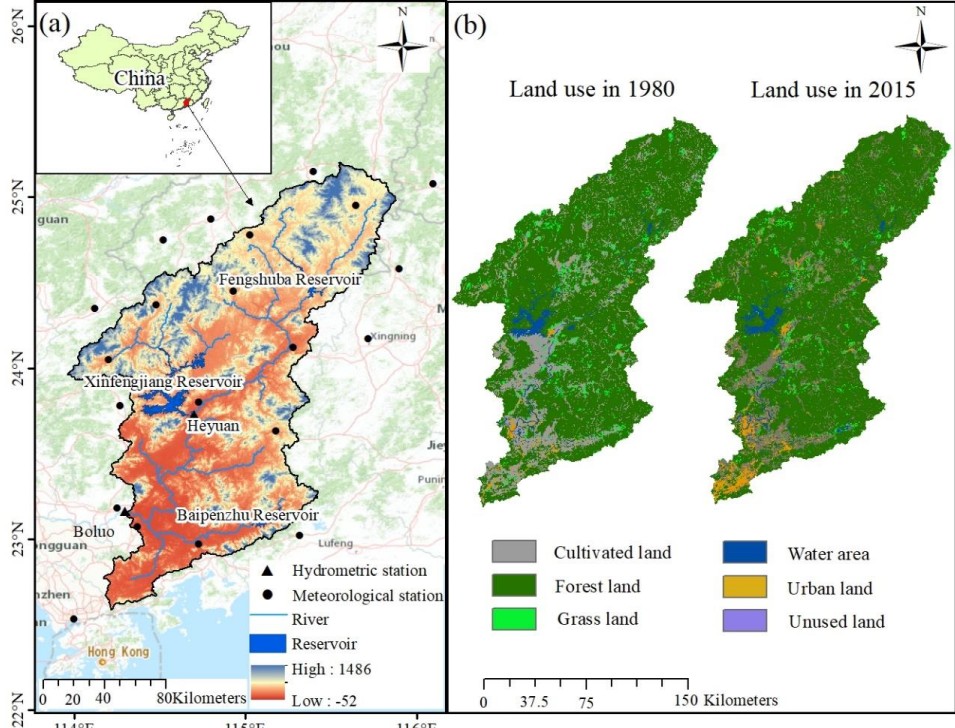


Figure 1. Location and characteristics of the study area: (a) location of the watershed, spatial distribution of the
hydrometeorological stations, and digital elevation model (Farr et al., 2007), (b) land use map (Xu et al.,

2018).

2.2 Methodology
2.2.1 SWAT model

The SWAT model was adopted to simulating hydrological processes and estimating the

volume of *BW* and *GW* for DRB (Arnold et al., 1998; Neitsch et al., 2002). The SWAT model is



widely applied to simulate streamflow and surface runoff (Arshad et al., 2022; Martínez-Salvador
and Conesa-García, 2020; Nie et al., 2023). The SWAT model is also widely utilized for exploring
the changes of *BW* and *GW* (Dai et al., 2022; Liang et al., 2018; Schuol et al., 2008).
In SWAT modelling, DRB was divided into 63 sub-basins, and each sub-basin was then
categorized into Hydrologic Response Units (HRUs) depending on land use, soils, and slope.
2.2.2 Model calibration and validation
In order to reduce the influence of hydraulic engineering, the SWAT model was calibrated
and validated utilizing monthly restored natural streamflow at the Boluo and Heyuan hydrometric
stations. The optimum hydrological parameters were shown in Table 1. All selected parameters are
automatically calibrated with 500 simulations via SWAT-CUP. The warm-up period for model
simulations is the first two years of the simulation period. Restored natural streamflow in 1970-
1979 was used to calibrate the model, and monthly time series of restored natural streamflow, *ET*
from GLEAM, and soil moisture from ERA5 during 1980-1989 were used to validate the model.
The calibration period for this research was 1970-1979, and the validation period was 1980-1989.
Three metrics, including the determination coefficient ($R^2$), the percentage bias (*PBIAS*), Nash-
Sutcliffe efficiency (*NSE*) were applied to evaluate simulation performance of SWAT model:
$$NSE = 1 - \frac{\sum_{i=1}^{n}(Q_{nat} - Q_{sim})^2}{\sum_{i=1}^{n}(Q_{nat} - Q_{ave})^2} \tag{1}$$





$$PBIAS = \frac{\overline{Q_{sim}} - Q_{ave}}{Q_{ave}} \times 100 \qquad (2)$$

$$R^2 = [\frac{\sum_{i=1}^{n}(Q_{nat} - Q_{ave})(Q_{sim} - \overline{Q_{sim}})}{\sqrt{\sum_{i=1}^{n}(Q_{nat} - Q_{ave})^2 \sum_{i=1}^{n}(Q_{sim} - \overline{Q_{sim}})}}]^2 \qquad (3)$$


where $Q_{nat}$, $Q_{ave}$, $Q_{sim}$, and $\overline{Q_{sim}}$ are monthly natural streamflow, mean monthly natural
streamflow, simulated streamflow, and mean monthly simulated streamflow, respectively, and $n$ is
the time step.

Table 1 Range of the main parameters and their optimal values in the calibration period

| Parameter | Calibration type | Initial range | Best calibrated value |
|---|---|---|---|
| GW_REVAP.gw | V | 0.19-0.2 | 0.199 |
| GWQMN.gw | V | 493-1247 | 916.493 |
| SLSUBBSN.hru | R | 2.6-5.7 | 2.804 |
| ESCO.hru | V | 0.89-0.97 | 0.901 |
| CN2.mgt | R | 0.14-0.27 | 0.209 |
| CH_K2.rte | V | 0.38-1.16 | 0.926 |
| ALPHA_BNK.rte | V | 0.12-0.18 | 0.165 |
| SOL_AWC.sol | R | 0.3-0.6 | 0.598 |
| SOL_K.sol | R | 0.32-0.69 | 0.669 |
| CH_K1.sub | V | 0-0.15 | 0.0295 |

Note: The symbols of V and R denote a replacement and a relative change to the default parameter value,
respectively.

This study reconstructed the natural monthly streamflow series of the basin by combining the

inflow and outflow of the three major reservoirs (Xinfengjiang Reservoir, Fengshuba Reservoir,
and Baipenzhu Reservoir) in DRB, based on the watershed water balance (Tu et al., 2018):
$$Q_{nat} = Q_o + \Delta Q = Q_o + Q_{in} - Q_{out} \qquad (4)$$





where $\Delta Q$ is the total reduced water volume, $Q_o$, $Q_{in}$, and $Q_{out}$ are the observed streamflow,
reservoir inflow, and reservoir outflow, respectively.

Overall, the SWAT model shows sufficient accuracies in simulating streamflow, actual

evapotranspiration, and soil moisture changes in DRB and can better simulate both seasonal and
interannual changes in streamflow. During the calibration period, both stations achieved $R^2$ above
0.9, $NSE$ exceeding 0.8, and $PBIAS$ less than 14% (Fig. 2). Both stations had simulation streamflow
$R^2$ greater than 0.8 during the validation period. The $NSE$ for streamflow simulation at the Heyuan
station and Boluo station of the validation were 0.81 and 0.74, respectively. The model performed
well in simulating the $ET$ and soil moisture. Since the GLEAM $ET$ data and ERA5 soil moisture
data are raster data of spatial resolution of 0.25×0.25°, considering the influence of data accuracy
on the results, this study uses the watershed scale to validate the simulation results of $ET$ and soil
moisture. In the validation period, the $R^2$ and $NSE$ for the simulation of evapotranspiration were
0.92 and 0.8, respectively (Fig. S1), while the $R^2$ and the $NSE$ for the soil moisture simulation were
both greater than 0.6. These validation results show that the model can be used to simulate
hydrological regimes in DRB.

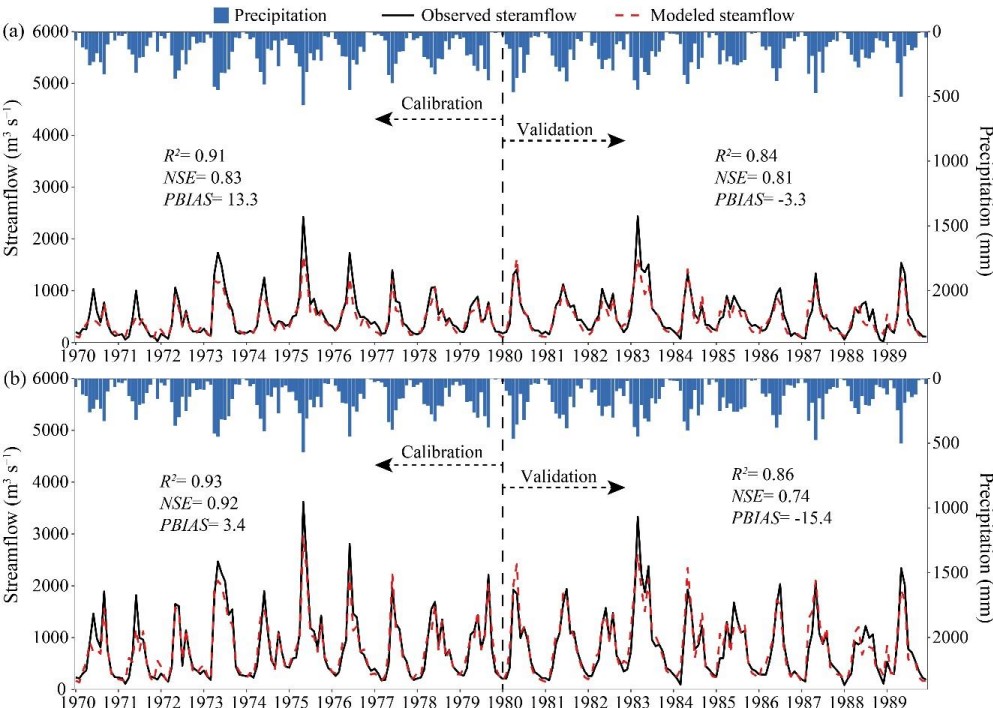

Figure 2. Simulated and observed monthly streamflow at the (a) Heyuan and (b) Boluo gauge stations during calibration and validation periods.

## 2.3 Scenario design and simulation

Three scenarios were formulated to assess the impacts of climate change and LUCC on *BW* and *GW* by changing climate conditions (land use) while holding land use (climate conditions) for each scenario simulation (Table 2). The LUCC for simulating the influences of climate change on hydrological cycle (S2-S1), while climate conditions change for simulating the influences of LUCC on hydrological cycle (S3-S2). The climate conditions and the land use were altered when assessing the joint influences of climate change and LUCC (S3-S1).

Table 2 Scenario settings for the simulation of effects of climate change and LUCC on blue and green water





| Scenarios | Land use | Climate period | Combined effects | Land use change effects | Climate change effects |
|-----------|----------|----------------|------------------|-------------------------|------------------------|
| S1 | 1980 | 1970-1993 | | | |
| S2 | 1980 | 1994-2017 | | | S2-S1 |
| S3 | 2015 | 1994-2017 | S3-S1 | S3-S2 | |

## 2.4 Calculation of blue and green water and water security indicators
### 2.4.1 Blue and green water scarcity
Blue water scarcity (*BWSC*) is determined by the quotient of *BW* withdrawal and availability.
The estimation of *BW* withdrawals (*BWR*) in this study involved the multiplication of the aggregate
population in each sub-basin by the combined water consumption per person (Liang et al., 2020).
The population of each sub-basin was extracted from the population raster data. *BW* availability
(*BWA*) represents the quantity of water that can be utilized without negatively impacting the river
ecosystems. Exhaustive exploitation of *BW* in rivers may adversely impacts river ecosystems.
Previous studies have generally used environmental flow requirements (*EFQ*) as a suitable metric
for sustaining robust ecosystems (Honrado et al., 2013). According to the study of Richter (2010)
and Richter et al. (2012), extracting more than 20% of the water from rivers may result in
ecological degradation. Therefore, 20% of streamflow can be deemed *BW* and used for water
supply. (Veettil and Mishra, 2016). The calculation of *EFR*, *BWA*, and *BWSC* are as follows:
$$EFQ_{(a,t)} = 0.8 \times Q_{\text{mean}(a,t)} \tag{6}$$
where $EFQ_{(a,t)}$ is the *EFQ* for sub-basin '*a*' during time '*t*'; $Q_{mean}$ is the long-term monthly average




streamflow.

$$BWA_{(a,t)} = Q_{(a,t)} - EFQ_{(a,t)} \qquad (7)$$

$$BWSC=BWR/BWA \qquad (8)$$

Green water scarcity (*GWSC*) is defined as the ratio between green water footprint (*GWFO*)
and green water availability (*GWA*). The *GWFO* denotes the actual evapotranspiration from the
watershed. *GWA* is the soil moisture that is available for evapotranspiration and vegetation
transpiration and is equal to the initial soil moisture (Liang et al., 2020). The *GWSC* can be
formulated as:

$$GWSC_{(a,t)}=GWFO_{(a,t)}/GWA_{(a,t)} \qquad (9)$$

where *GWSC* is green water scarcity;*GWFO*$_{(x,t)}$ is the actual evapotranspiration;*GWA*$_{(a,t)}$ is initial
soil moisture。
2.4.2 Regional water stress
The Falkenmark index (*FLK*) (Falkenmark et al., 1989) is a widely used measures of water
stress, defined as the proportion of *BWA* to the overall population. The Falkenmark index is
classified into no stress, stress, scarcity, and absolutely scarcity based on per capita water use.
Absolute scarcity is regarded to occur in areas where the indicator threshold is less than 500 $m^3$
capita$^{-1}$ a$^{-1}$, and no stress is thought to occur in areas where the threshold is larger than 1700 $m^3$
capita$^{-1}$ a$^{-1}$.


### 2.5 Calculation of relative contribution

The influences of climate change and LUCC on the changes of blue and green water in different periods are evaluated utilizing the relative contribution rate in this research (Li et al., 2021):

Climate change contribution to *BW* and *GW* change is estimated by:

$$CR_C = \frac{|X_2 - X_1|}{|X_2 - X_1| + |X_3 - X_2|} \times 100\%$$

(10)

where *X* is the amount of *BW* or *GWF* and *GWS* for each scenario.

The contribution of LUCC to *BW* and *GW* change is estimated by:

$$CR_L = \frac{|X_3 - X_2|}{|X_3 - X_2| + |X_2 - X_1|} \times 100\%$$

(11)

### 2.6 Data

The dataset used in this study consists of three parts: (1) hydrometeorological data, (2) geospatial data encompassing DEM, soil type, and land use, and (3) socioeconomic data encompassing per capita water consumption and population data.

Observed monthly streamflow data of the two hydrological stations in the study were collected for the years 1970-2000 from Boluo Station and Heyuan Station, and the observed streamflow time series of these two hydrological stations are of no missing data. Monthly inflow and outflow data of the three major reservoirs in DRB were also collected. All hydrologic data



were obtained from the Guangdong Provincial Hydrological Bureau. Meteorological data of daily
precipitation, temperature, and other meteorological data for 1968-2017 from 21 Meteorological
stations in the watershed were obtained from the National Meteorological Information Center of
the China Meteorological Administration. Monthly actual *ET* data for SWAT model validation was
obtained from the Amsterdam Evapotranspiration Model dataset with a spatial resolution of 0.25°
× 0.25° (Martens et al., 2017). Monthly soil moisture data for SWAT model validation was obtained
from the European Center for Medium-Range Weather Forecasts ERA5-land dataset with a spatial
resolution of 0.1° × 0.1° (Muñoz Sabater, 2019). The actual evapotranspiration and soil moisture
of the watershed equals to the average of all grids included in DRB.

The 90 meter resolution DEM data and 30 meter resolution land use data at ten-year intervals

(i.e., 1980, 1990, 2000, 2010, 2015) are obtained from the Data Center for Resources and
Environmental Sciences of the Chinese Academy of Sciences (Xu et al., 2018). Soil data is
obtained from the 1-km resolution Harmonized World Soil Database dataset from the Food and
Agriculture Organization of the United Nations (Fischer et al., 2008).

The annual per capita integrated water consumption data of DRB from 2000-2017 was

acquired from the Water Resources Bulletin of Guangdong Province. The population data in 2000,
2005, 2010, and 2015 was obtained from the 1 × 1 km spatial raster data of the Resource and
Environment Science and Data Center of the Chinese Academy of Sciences (Xu, 2017).





## 3 Results

### 3.1 LUCC and Climate variability in DRB

LUCC in DRB is mainly the decrease of cultivated land and the increase of urban land. The land use in DRB primarily consisted of forest land (18,875-18833 km$^2$), which is more than 70% of DRB. From 1980 to 2015, the urban land and water areas showed an increase of 469.4 km$^2$ (137%) and 17.4 km$^2$ (2.8%), while the grass land, cultivated land, and forest land showed a decrease of 41.3 (4.3%), 487.5 (10.8%), and 42.1 km$^2$ (0.2%), respectively (Table 3).

Table 3 Land use transfer matrix in DRB from 1980 to 2015

| Land use type | | Grass Land (km$^2$) | Urban land (km$^2$) | Cultivated Land (km$^2$) | Forest land (km$^2$) | Water area (km$^2$) | Unused land (km$^2$) | 1980 total (km$^2$) |
|---|---|---|---|---|---|---|---|---|
| | | **2015** | | | | | | **1980** |
| 1980 | Grass land | 795.6 | 29.9 | 18.3 | 123.5 | 2.5 | 0.0 | 969.7 |
| | Urban land | 0.6 | 319.6 | 12.4 | 7.6 | 2.3 | 0.0 | 342.4 |
| | Cultivated land | 19.0 | 269.8 | 3771.7 | 427.9 | 40.4 | 0.03 | 4528.8 |
| | Forest land | 110.7 | 183.7 | 226.2 | 18278.7 | 33.1 | 0.02 | 18832.5 |
| | Water area | 2.5 | 8.9 | 12.7 | 36.8 | 551.0 | 0.00 | 611.9 |
| | Unused land | 0.0 | 0.0 | 0.02 | 0.03 | 0.00 | 0.45 | 0.51 |
| | 2015 total | 928.4 | 811.9 | 4041.3 | 18874.5 | 629.2 | 0.51 | 25285.8 |

DRB exhibited significant regional differences in multi-year average precipitation, temperature, and potential evapotranspiration. The precipitation exhibited an increasing trend from the central to the south and north of DRB. The temperature and potential evapotranspiration showed an overall distribution pattern of greater values in the south and minor values in the north





of DRB (Fig. 3). The multi-year average precipitation for the entire of DRB was 1790.1 mm, with
annual precipitation ranging from 1236.2-2567.5 mm. The regions with the highest multi-year
average annual precipitation are located in the southeast of DRB, where annual precipitation
exceeds 2200 mm, while the regions with the lowest precipitation are in the northeastern of the
watershed. The average annual temperature in DRB ranged from 19.5-21.3 °C, and the average
annual potential evapotranspiration ranged from 1101.5-1320.6 mm. The south of DRB is
predominantly urban, characterized by the urban heat island effect, while the north of DRB is
mountainous with higher elevations, leading to the spatial distribution of temperatures.

The average temperature and potential evapotranspiration at DRB meteorological stations

exhibited significant variations, while precipitation showed a relatively minor trend (Fig. 3).
Overall, basin-averaged precipitation and potential evapotranspiration showed a non-significant
decreasing trend, while temperatures showed a significant increasing trend. There was no
significant change trend of precipitation for all stations in DRB (Fig. 3a). Twenty out of 21
meteorological stations in the region showed statistically significant increasing trends in average
temperature, indicating a warming trend (Fig. 3b). Nine stations showed a significant decreasing
trend in potential evapotranspiration, primarily located in northern DRB (Fig. 3c).



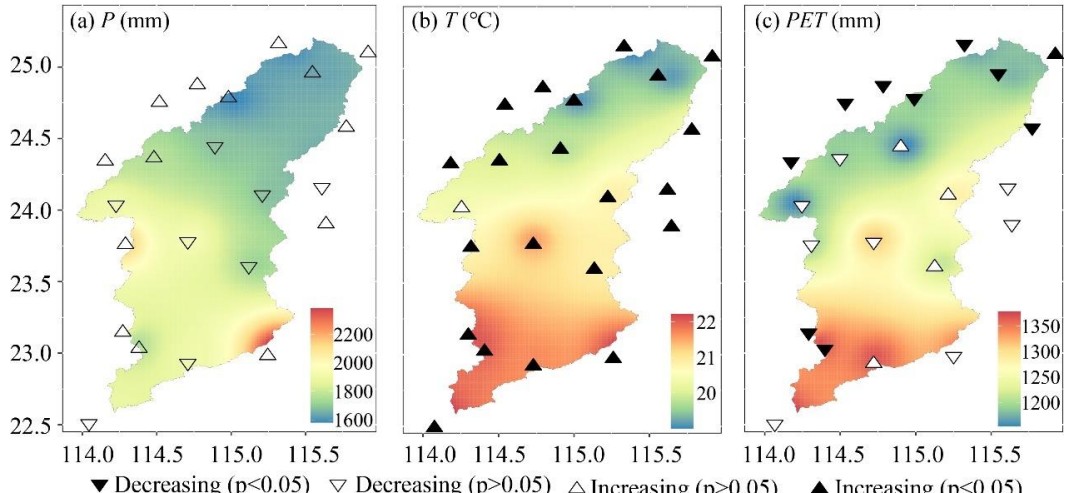

Figure 3. Spatial distribution of annual mean (a) precipitation, (b) temperature, (c) potential
evapotranspiration in DRB from 1960-2017. Each triangle represents the Mann-Kendall test result at a
meteorological station.
## 3.2 Blue and green water resources
The average annual $BW$ and $GW$ were 1240.8 and 840.7 mm, respectively. The DRB water
resources were dominated by $BW$, representing 60.1% of the total water resources, and $BW$ was
1.48 times higher than that of $GW$ resources. The average $GWF$ and $GWS$ were 689.3 and 151.4
mm, respectively.
The annual $BW$ resources in the sub-basins of DRB ranged from 893.7-1990 mm, showing
an increasing trend from the central to the south and north of DRB, aligning with the spatial
distribution of precipitation (Fig. 4a). The regions with abundant $BW$ resources are situated in the
central and southeast parts of DRB (>1300 mm), and the $BW$ in the upper reaches is comparatively
low (<1100 mm). Differences in the spatial distribution of $BW$ are primarily caused by differences



in the spatial distribution of precipitation. Overall, the *GWF* and *GWS* are more evenly distributed
in the sub-basins than *BW*. The annual *GWF* in the sub-basins of DRB ranged from 573.6-923.6
mm. The sub-basins with higher *GWF* are primarily located in the Xinfengjiang reservoir area in
the middle reaches (>700 mm), while the low *GWF* sub-basins are situated in the southwest of
DRB (<600 mm) (Fig. 4b). The land use in the sub-basins where Xinfengjiang Reservoir is located
is primarily water areas, with a higher water evaporation rate than other regions, resulting in a
greater *GWF* in this area than in other regions. The annual *GWS* in the sub-basins of DRB ranged
from 126-190.6 mm. The sub-basins with higher *GWS* are mainly located in the lower part of DRB
(>150 mm) (Fig. 4c). The distribution pattern of *GWS* resources has a great relationship with the
soil type of the watershed. The upper reaches and the northwestern part of the watershed are mostly
red soil, while the middle and lower reaches are dominated by reddish soil. Reddish soil has a
smaller water storage capacity than red soil, loses water faster, and has weaker water conservation
and water supply performance than red soil. This is the primary factor for the north-south
discrepancies in the amount of *GWS* resources in DRB. In addition, the southern region is mostly
of large and medium-sized cities. As urban construction land expands, the land use type in the
region has gradually changed to urban land, industrial land, etc., and the solidification of road
surfaces has reduced the area of bare soil in the region, resulting in a decrease in *GWS* resources.
The annual *GWI* (Fig. 4d) showed a spatial pattern opposite to *BW*, decreasing from 0.45 in the
upper reaches to 0.3 in the lower reaches. The highest *GWI* is found in the upper reaches, which is
due to the relatively low rainfall in the upper reaches and the lush vegetation, with significant plant
interception and transpiration, resulting in a higher proportion of total evapotranspiration than in
the middle and lower reaches. The central part of the basin has the highest precipitation, leading
to a lower *GWI*. The southern part of the watershed has the highest temperature, and
evapotranspiration is high. Meanwhile, the lower reaches have a large proportion of agricultural
and urban land, and crop irrigation can increase evapotranspiration.

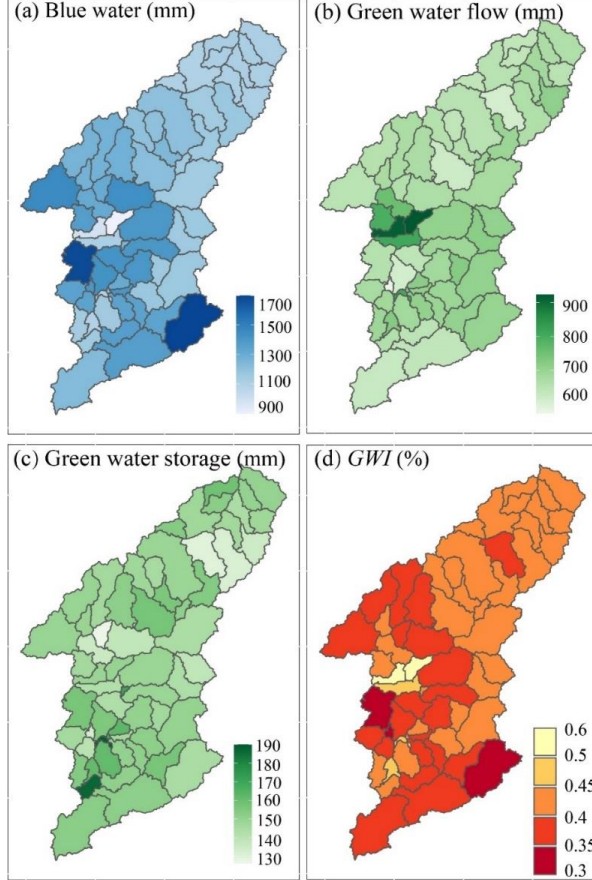


Figure 4. Spatial distribution of mean (a) *BW*, (b) *GWF*, (c) *GWS*, (d) *GWI* in DRB over during 1970-2017.
In DRB, there was no significant increasing trend in either BW or GWS, while GWF



exhibited a significant decreasing trend. The annual trend rate of BW in DRB was 0.14 mm a-1,
with an annual fluctuation range of 713.6-2017.5 mm during 1970-2017. The minimum BW
occurred in 1991, while the maximum was recorded in 2016 (Fig. 5a). The GWF in DRB from
1970 to 2017 exhibited a significant decreasing trend (-0.57 mm a-1) (Fig. 5b). The minimum
GWF occurred in 2005 (603.1 mm), while the maximum was recorded in 1974 (721.3 mm). In
contrast, the GWS in DRB from 1970 to 2017 has been slowly increasing at a rate of 0.015 mm a-
1 (Fig. 5c). The annual fluctuation in GWS was smaller than BW and GWF. The GWI in DRB
from 1970 to 2017 showed no significant decreasing trend at a rate of -0.0003 % a-1 (p>0.05) (Fig.
5d), implying that the redistribution of precipitation in DRB might change slowly.

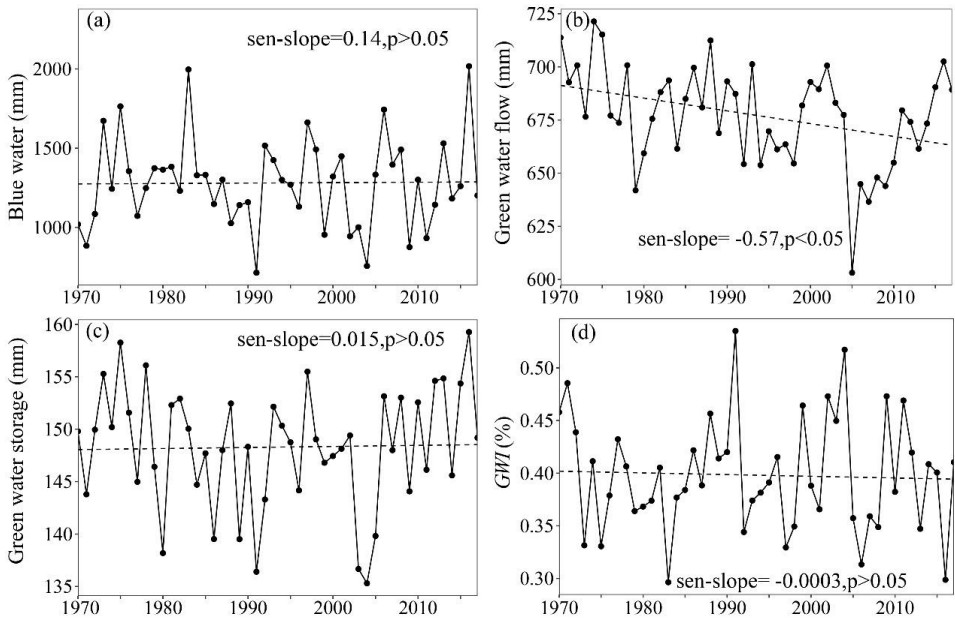


Figure 5. Interannual variation of (a) *BW*, (b) *GWF*, (c) *GWS*, (d) *GWI* in DRB during 1970-2017.





## 3.3 Blue and green water scarcity

The average blue water scarcity level in DRB was low (22.4%) during 1970-2017. The blue

water scarcity levels in various sub-basins ranged from 0.1-206%. The multi-year average blue

water scarcity, except for one sub-basin in the southwest, was all low (<100%) (Fig. 6a). This

indicates that blue water scarcity is not common in DRB at the annual scale. Regions with

relatively high blue water scarcity (>20%) are mostly situated in the upper reaches of various

tributaries within the watershed, where river streamflow is relatively small. The area with the

highest blue water scarcity (206%) is located in the 63rd sub-basin of Shenzhen and Huizhou,

reaching a moderate level of blue water scarcity. This region has a large population, with a much

higher blue water demand than other areas. Additionally, this sub-basin is situated in the upper

reaches of the primary tributary of DRB, resulting in a limited supply of $BW$ resources. Although

the northern parts of sub-basins 55, 56, and 61 have large populations, these sub-basins are situated

in the downstream of the main Dongjiang River, with a higher streamflow, leading to lower $BWSC$

levels. The average $GWSC$ in the entire basin from 1970-2017 was low (41.4%). The blue water

scarcity levels in various sub-basins ranged from 31-104%. The vegetation cover in DRB is high,

and DRB is thus of relatively high rates of vegetation transpiration and interception evaporation.

The basin experiences a $GWSC$ of nearly 50%, indicating a potential occurrence of $GWSC$. The

areas with higher $GWSC$ are primarily situated in the middle reaches for DRB (Fig. 6b), where

water surface evaporation is high, resulting in their $GWSC$ exceeding 100%. The evaporated water



in these areas originates from the reservoirs, not the soil, leading to an overestimation of the *GWSC*
in these sub-basins.

Furthermore, the *FLK* index was also used to quantify population-driven water resource

scarcity. F1-F4 represent absolute scarcity, scarcity, stress, and no stress, respectively. The results
showed that most regions in DRB have no water scarcity pressure (Fig. 6c). However, the 63rd
sub-basin experienced absolute water scarcity, and the 52nd sub-basin experienced water scarcity.
There were six lower reaches sub-basins and four upper reaches sub-basins facing water stress.
DRB receives ample precipitation, resulting in a relatively large river flow, generally leading to a
higher *FLK* index. As a result, the basin faces lower water resource pressure.

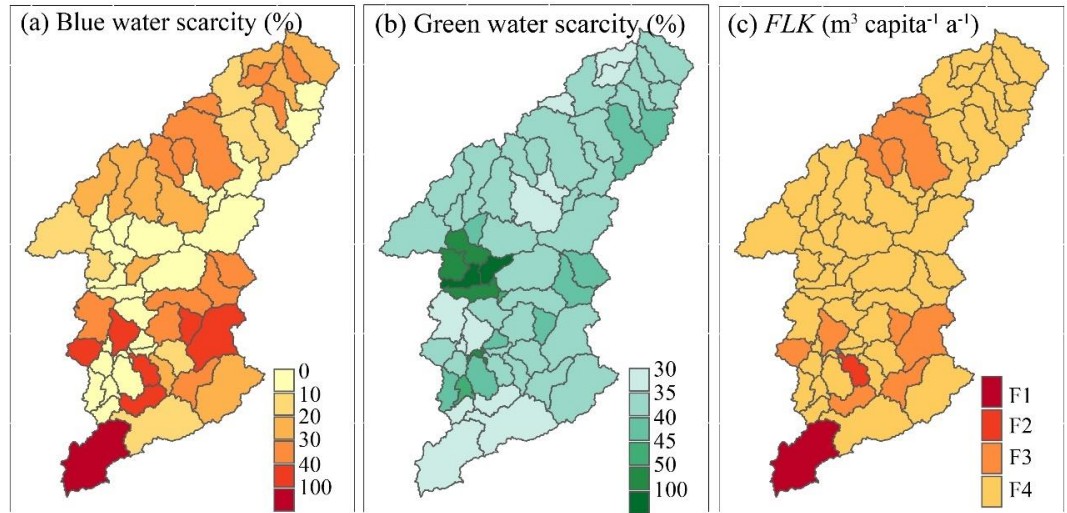

Figure 6. Spatial distribution of mean (a) *BWSC*, (b) *GWSC*, and (c) *FLK* index in DRB over during 1970-
2017.

This study also further identified hotspots of *BWSC* and *GWSC* in DRB by hierarchical

clustering of *BWSC* and *GWSC* in each sub-basin. Figure 7 shows the clustering tree results for





*BWSC* and *GWSC*. When the standardized distance was set to 500, all sub-basins could be divided
into four categories according to blue water scarcity: (1) The first category consisted of 27 sub-
basins such as 32, 56, and 28, where the blue water scarcity level was the lowest (<20%). (2) The
second category comprised sub-basin 63, which has the most severe blue water scarcity (206%).
(3) The third category comprised seven sub-basins, such as 52, 58, and 60, all located in the lower
reaches, with relatively high blue water scarcity levels (40%-100%). These sub-basins are mostly
located in the tributaries of lower reaches, with a relatively large population and smaller river
streamflow compared to the mainstem of the Dongjiang River. (4) The fourth category consisted
of 28 sub-basins, such as 59, 62, and 8, with blue water scarcity levels ranging from 20% to 40%.
Similarly, hierarchical clustering was conducted for *GWSC*. When the standardized distance was
set to 500, *GWSC* in the sub-basins could be divided into three categories: (1) The first category
consisted of 56 sub-basins, such as 37, 56, and 29, with relatively low *GWSC* levels, all below
50%, indicating low *GWSC*. (2) The second category consisted of sub-basins 32 and 33, where the
predominant land use type was water areas, leading to higher *GWSC* due to high water surface
evaporation. (3) The third category consisted of sub-basins 47, 31, 54, 30, and 36, where the water
area proportion in these sub-basins was larger than in others, leading to significant influences from
water surface evaporation.



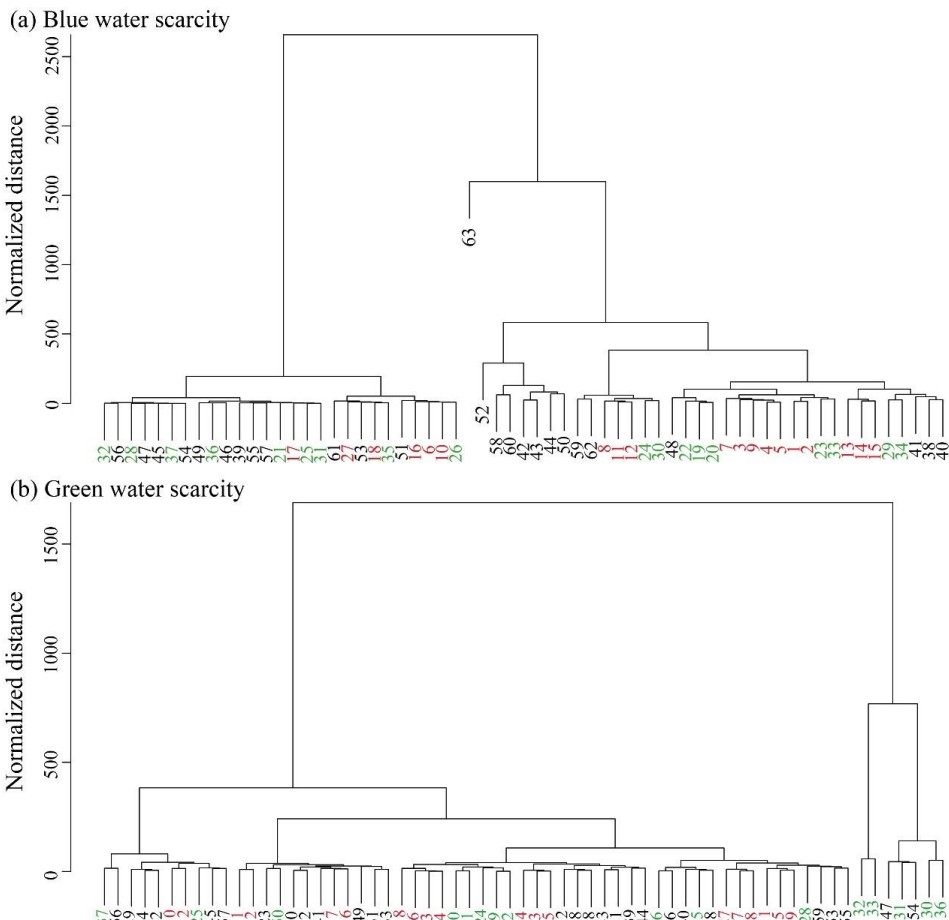

Figure 7. Hierarchical clustering tree of (a) *BWSC*, (b) *GWSC*.

The interannual variations in *BWSC* and *GWSC* in DRB showed distinct regional differences. *BWSC* in the basin was slowly increasing at a rate of 0.3% a$^{-1}$ (Fig. 8a). The *BWSC* in lower reaches slowly increased at a rate of 1.1 % a$^{-1}$, while the *BWSC* in upper and middle reaches slowly decreased at -0.47% a$^{-1}$ and -0.1% a$^{-1}$, respectively. *GWSC* in the upper, middle and lower reaches of DRB showed a decreasing trend, with basin scale *GWSC* decreasing significantly at a rate of -0.04% a$^{-1}$ (Fig. 8b). Despite the acceleration of urbanization and a significant increase in


population in the middle and lower reaches of the watershed, blue water availability and the
amount of obtainable *BW* have been increasing. Additionally, the annual per capita water
consumption in the basin has decreased from 481.0 m³ in 2000 to 245.0 m³ in 2020. As a result,
the rate of increase in *BWSC* in the watershed has been relatively small. In contrast, the *GWF* in
DRB demonstrated a significant decreasing trend, and the *GWS* increased slowly. Therefore, the
*GWSC* in DRB demonstrated a significant decreasing trend. Meanwhile, the *FLK* index of the
watershed showed a significant decreasing trend (-285.3 m³ per year), which means that the per
capita water resources in the watershed have significantly decreased (Fig. 8c). This is due to the
rapid population growth in the watershed and the slow increase in available water resources.

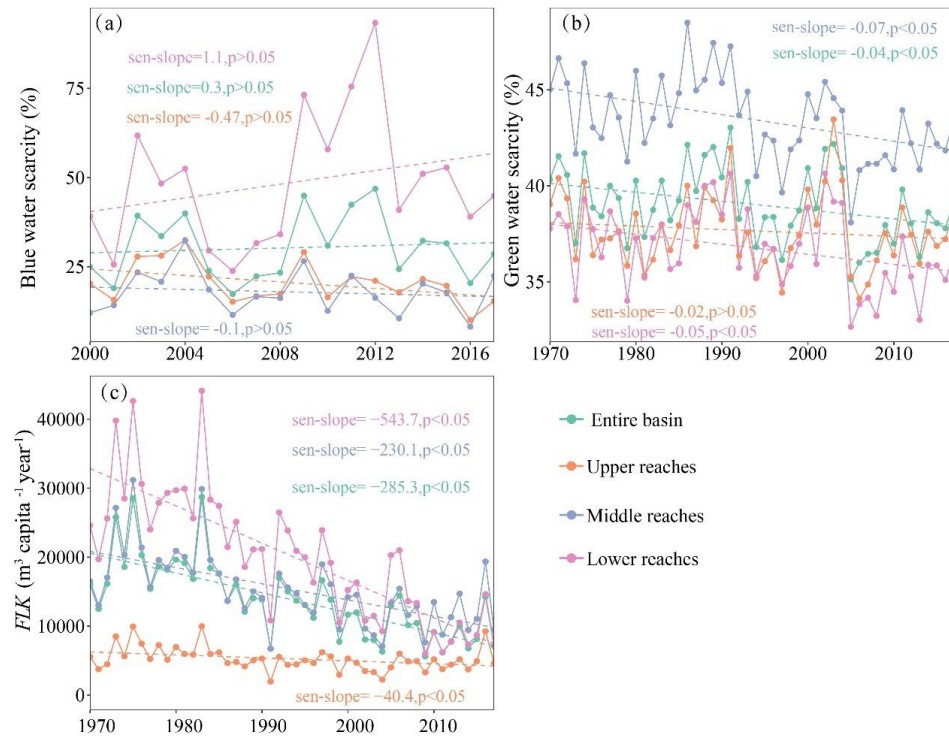


Figure 8. Interannual variation of(a) *BWSC*, (b) *GWSC*, and (c) *FLK* index in DRB during 1970-2017.





### 3.4 Impacts of LUCC and climate change on blue and green water


To examine the impacts of climate change and LUCC on *BW* and *GW* change, this study set
three climate conditions and land use scenarios to explore this effect by comparing the scenarios
(Table 3). The combined impacts of climate change and LUCC on *BW* and *GWS* in DRB were
superimposed, and the combined effect on *GWF* was a negatively synergistic effect. Figure 6
shows the variations in *BW* and *GW* under the impacts of climate change (S2-S1) and LUCC (S3-
S2), as well as their combined effects (S3-S1), along with the relative contribution of climate
change and LUCC to the *BW* and *GW* changes in DRB during 1970-2017. Under the joint
influences of climate change and LUCC, *BW* decreased by 4.5 mm a$^{-1}$. Among this decrease,
climate change resulted in a loss in *BW* of 3.9 mm a$^{-1}$, contributing 88.0%, while LUCC led to a
loss in *BW* of 0.5 mm a$^{-1}$, contributing 12.0% (Fig. 9a). The effect of climate change on *BW*
variation is much greater than that of LUCC at the basin scale. Under the combined influences of
climate change and LUCC, *GWF* decreased by 17.0 mm a$^{-1}$. Among this decrease, climate change
accounted for a decrease in *GWF* of 19.5 mm a$^{-1}$, contributing 88.5% to the decrease, while LUCC
led to an increase in *GWF* of 2.5 mm a$^{-1}$, contributing 11.5% (Fig. 9b). Overall, the influence of
climate change on *GWF* changes in the watershed is significantly more pronounced than that of
LUCC. Under the joint influences of climate change and LUCC, *GWS* increased by 0.7 mm a$^{-1}$.
Among this increase, climate change contributed to an increase in *GWS* of 0.3 mm a$^{-1}$, accounting
for 39.4%, while LUCC contributed to an increase in *GWS* of 0.4 mm a$^{-1}$, accounting for 60.6%
(Fig. 9c). DRB is situated in a humid regions with high *GWS*, resulting in small fluctuations of
*GWS* in response to precipitation changes. The fluctuations of *GWS* are primarily influenced by
the soil properties and land use. In general, the effect of climate change on the *GWS* change of
DRB is smaller than the effect of LUCC.

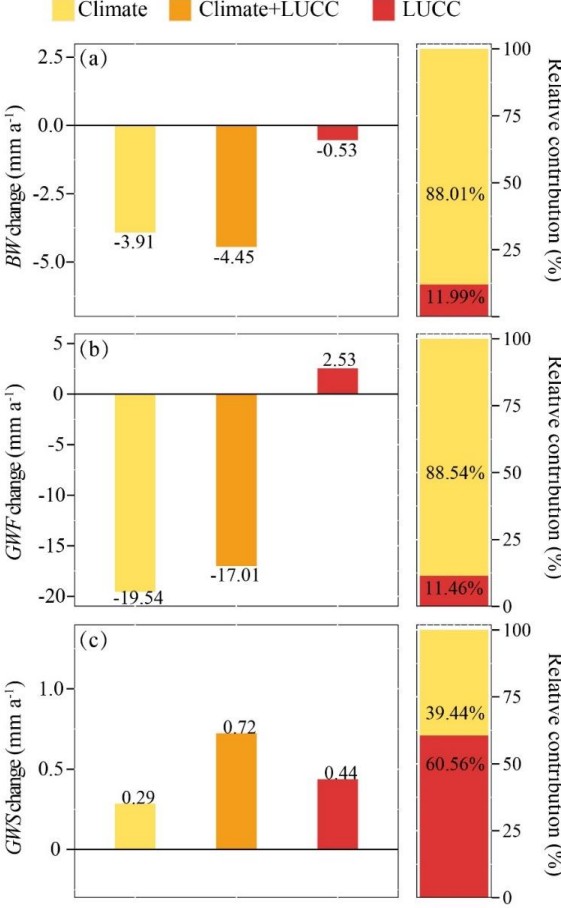


Figure 9. Effects and relative contribution of climate change and LUCC on the changes in (a) *BW*, (b) *GWF*,
(c) *GWS* in DRB during 1970 to 2017.
Under the coupled influences of climate change and LUCC, the *BW* and *GW* resources in
DRB have changed. However, there were differences in the joint impacts of climate change and



LUCC on *BW* and *GW*. Both climate change and LUCC have led to the decrease of *BW* in the
watershed, and the combined effect of climate change and LUCC on *BW* equals to the sum of their
individual effects. Climate change, such as a decrease in potential evapotranspiration, has resulted
in a decrease in *GWF* in DRB, while LUCC has led to an increase in *GWF*. Therefore, the joint
impacts of climate change and LUCC on *GWF* was partially offset, resulting in the joint impacts
of climate change and LUCC on *GWF* being less than the sum of their absolute individual effects.
Both climate change and LUCC have led to an increase in *GWS* in DRB, and the joint impacts of
climate change and LUCC on *GWS* equals to the sum of their individual effects.

## 4 Discussion

This study used the SWAT model to simulate the changes in *BW* and *GW* resources in DRB
over the past five decades and their response to climate change and LUCC. It also assessed the
water resource security in the basin. The results indicate that the total water resources showed a
decreasing trend in the past five decades in the entire DRB mainly due to decreases in precipitation,
which is similar to what Zhu et al. (2022) found. The findings also revealed that the *GWF* exhibited
a decreasing trend, and the *BW* and *GWS* exhibited an increasing trend. Liu et al. (2010) similarly
found an increasing trend in annual surface runoff in DRB. Potential evapotranspiration and solar
radiation in DRB showed a decreasing trend, which may be the main cause of the significant
decrease in *GWF* in the basin (Fig. S2), and similar conclusions are obtained in He et al. (2013).



We show that water resources in DRB are dominated by *BW*, with a mean annual *GWI* of 0.4,
which is the same as what many studies show in humid areas (Nie et al., 2023). Although the *GWI*
in humid areas is much smaller than that in arid areas, the ratio of *GW* in DRB still reach 40%, so
it is imperative to incorporate *GW* in the water resources assessment system. The *GWI* in the upper
and middle reaches of DRB exceeded 0.4, while that in the lower reaches was only about 0.3.
These results mean that to ensure the appropriate utilization of water resources, effective water
management in the upper and middle reaches of DRB should consider *GW* planning while water
management in the lower reaches should mainly consider *BW*. The assessment results of *BWSC*
and *GWSC* in DRB similarly illustrates this issue. The *GWSC* in the upper and middle reaches was
bigger than that in the lower reaches of DRB, while the *BWSC* in the lower reaches of DRB was
bigger than in the upper and middle reaches (Fig. 8).
There are robust correlations between *BW* and precipitation, *GWF* and potential
evapotranspiration in DRB. Climate change plays a dominant role in variations of *BW* and *GWF*.
*BW* is more sensitive to precipitation and potential evapotranspiration. *GWF* shows sensitivity to
changes in potential evapotranspiration and *GWS* is influenced by both precipitation and potential
evapotranspiration (He et al., 2015; Jeyrani et al., 2021). Of course, there are also studies for arid
regions show that *GWF* is mainly affected by precipitation (Jun Wu et al., 2021), which may be
linked to the hydrothermal conditions of the basin. There is sufficient precipitation in DRB, where
the *GWF* changes are mainly energy-limited, and effect of precipitation on the *GWF* are smaller.





Although *BW* and *GW* are mainly affected by climate change, the influences of LUCC on

them cannot be ignored. The reaction of water resources to LUCC is exceedingly intricate and

involves various hydrological processes, including runoff yield, infiltration, and groundwater (Cuo,

2016; Zhang and Shangguan, 2016). As there is a strong compensatory effect of diverse land use

in the hydrological system, particularly in expansive watersheds, this could create a strong

resistance to *GW* and *BW* conversion (Lin et al., 2015). Decrease in forest land or increase in

cultivated and urban land could lead to an rise in *BW* and a decline in *GW* in the watershed. Veettil

and Mishra (2018) demonstrate that there is a 10% rise in forest land cover and a 1.4% drop in *BW*,

indicating a negative elasticity between the two. However, the effect of urban land on streamflow

in different time periods showed the opposite effect. On the one hand, the increase in urban land

results in increases in impermeable area and thus surface runoff in the basin, but at the same time,

the increase in urban land may also reduce groundwater discharge to streamflow. At the same time,

LUCC often results in changes in vegetation. Vegetation variations affect the water cycle by

altering canopy interception (Shao et al., 2018; Jianping Wu et al., 2019), transpiration (Chen et

al., 2023) and canopy evaporation, and ameliorating soil structure (Qiu et al., 2022), Thus

increasing vegetation often increases infiltration and soil moisture and reduces surface runoff.

There are several limitations and uncertainties in this research. (1) Since the quantity of the

*BW* and *GW* is derived from the output results of the model simulations, including water yield, *ET*,

soil moisture, and groundwater, the precision of the outcomes depends largely on the precision of



the model simulations. Given the absence of observed evapotranspiration and soil moisture data
for DRB, this study calibrated and validated the SWAT model using only monthly streamflow,
which may weaken these results to some extent. To enhance the credibility of the model, this study
also utilized widely used actual evapotranspiration data (GLEAM) and soil moisture (ERA5-land)
during model validation at a basin scale. The findings indicated that the simulation performance is
relatively good and meets the accuracy requirements for simulation. (2) Climate change, LUCC,
and large reservoir operation are the primary factors influencing the changes in hydrological
conditions in DRB. The contributions of reservoir regulation, LUCC, water resource utilization,
and climate change to the distribution of intra-annual flow are 33.5%, -9%, 4.5%, and 1%,
respectively, during 1956-2009 (Tu et al., 2015). The operation of reservoirs, including large
reservoirs like the Xinfengjiang Reservoir, is one of the important reasons for hydrological changes
in DRB (Lin et al., 2014; Zhang et al., 2015). The reservoir module was not established when
constructing the SWAT model in this research. To obtain natural *BW* and *GW* volumes in the
watershed and mitigate the impact of hydraulic engineering, reconstructed natural streamflow
based on observed flow was utilized for model calibration and validation. However, hydraulic
engineering significantly influences the annual allocation of *BW*. The flow restoration considered
the impacts of the three major reservoirs on the Dongjiang River and did not consider the impacts
of other minor hydraulic projects and human water consumption. (3) Both the calculations of
*BWSC* and the *FLK* index include environmental flows. This study represented the proportion of



environmental flow in streamflow as 80%. Some studies have suggested that assuming
environmental flow to be 80% of the total water resources in a basin may overestimate water
scarcity (Liu et al., 2017; Richter et al., 2012). Therefore, we varied the proportion of
environmental flow and assessed the degree of *BWSC* using 60% and 70% proportions. Results
show that only the 63rd sub-basin changed from severe *BWSC* to moderate to high *BWSC*, while
other sub-basins remained with low *BWSC*. Therefore, the threshold for environmental flow has a
minor impact on this paper. The assessment of *BWSC* and per capita water resources did not take
into account the water demand of cities such as Shenzhen and Hong Kong, although the water
supply for these cities primarily comes from the Dongjiang River through the Dongjiang-Shenzhen
Water Supply Project. (4) The hydrological modeling approach utilized in this research is a
frequently used method for quantitative analysis of attribution. Nevertheless, it implies
independence between climate change and LUCC and does not adequately distinguish the impacts
of these two components. Such restriction is diffusely recognized to exist (Dey and Mishra, 2017).
Despite this recognized limitation, hydrological modeling methods have been widely used in
numerous similar researches, yielding credible results (Li et al., 2021; Nie et al., 2023).

**5 Conclusion**


This study analyzed the spatio-temporal evolution of *BW* and *GW*, assessed the water security,
and evaluated the effects of climate change and LUCC on *BW* and *GW* in DRB using the SWAT





model. The conclusions can be outlined as follows:

(1) During 1970-2017, grass land, cultivated land, and forestland in DRB decreased by 4.3%,

10.8%, and 0.2%, respectively, while urban land and water areas increased by 137% and 2.8%,
respectively. The annual precipitation and potential evapotranspiration showed a non-significant
decreasing trend, while the annual average temperature showed a significantly increasing trend.

(2) The annual *BW*, *GWF*, and green storage in DRB from 1970-2017 were 1240.8 mm, 840.7

mm, and 151.4mm, respectively. *BW* (0.14 mm a$^{-1}$) and *GWS* (0.015 mm a$^{-1}$) in DRB showed no
significant increasing trend, and *GWF* (-0.57 mm a$^{-1}$) showed a significant decreasing trend.

(3) The level of annual *BWSC* and *GWSC* in DRB were low, and per capita water resources

exceeded 1,700 m$^3$ capita$^{-1}$ a$^{-1}$. *BWSC* displayed a non-significant increasing trend, while the
*GWSC* and *FLK* index displayed a significant decreasing trend, especially in lower reaches.

(4) Climate change was the major driving factor of changes in *BW* and *GWF*, and LUCC was

the major driving factor of *GWS* change. Climate change contributed to88.0%, 88.5%, and 39.4%
of the changes in *BW*, *GWF*, and *GWS* in DRB, respectively. Both climate change and LUCC
decrease *BW*, while climate change (LUCC) decrease (increase) *GW* flow in DRB.
**Competing interests**

The contact author has declared that none of the authors has any competing interests.



**Acknowledgments**

This study was supported by the National Key Research and Development Program of China

(2021YFC3001000), the Science and Technology Innovation Program from Water Resources of
Guangdong Province (2023-01), and the National Natural Science Foundation of China (52179029,

52179030).



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
