# Peer review of "Combined impacts of climate change and human activities on blue and green water resources in the high-intensity development watershed"

_Hydrology and Earth System Sciences, 2024_

## Author Comment (AC1)

*The manuscript "Combined Impacts of Climate Change and Human Activities on Blue and Green Water Resources in the High-Intensity Development Watershed" presents a comprehensive and insightful study on the variations in blue water (BW) and green water (GW) resources in the Dongjiang River Basin (DRB). The use of the Soil and Water Assessment Tool (SWAT) model to quantify the impacts of climate change and land use change (LUCC) on BW and GW provides robust and valuable findings. The study's relevance to the Guangdong-Hong Kong-Macao Greater Bay Area (GBA) underscores its importance in guiding sustainable water resource management in a rapidly developing region. However, several issues need addressing to enhance the paper's clarity and impact.*

We wish to sincerely thank the reviewers for their extensive and thoughtful comments on our manuscript which we have addressed in the revised manuscript as discussed below. Throughout, reviewer comments are in *blue* font and *italic* type, and **our response** in **black** font.

*Major Concerns:*

*1. Formulation and Demonstration of GW and BW Derivation: The paper should clearly formulate and demonstrate how GW and BW are derived using the SWAT model. Without this critical information, the reader's understanding of the methodology and results is hindered. Additionally, for equations 10-11, it is necessary to specify clearly what "X" represents. This clarification is essential for comprehending these equations fully.*

Response: We have added the formulation and demonstration of *GW* and *BW* derivation. Lines 205-214 in the revised manuscript:

*2.4.1 Calculation of blue and green water*

*BW is calculated from the sum of water yield (SWAT output WYLD) and groundwater storage. The former refers to the amount of water that leaves the HRU*

*and enters the channel. The latter represents the net amount of water recharged to aquifers (SWAT output GW_RCHG) and the amount of aquifer water discharges to the main channel (SWAT output GW_W) during a time step (Hordofa et al., 2023). GW can be divided into two components including GWF which is the actual evapotranspiration (SWAT output ET) from the HRU, and GWS which is the soil water moisture (SWAT output SW) (Nie et al., 2023; Veettil and Mishra, 2018). The calculation of the Green Water Index (GWI) involves dividing the quantity of GW by the sum of BW and GW (Ding et al., 2024; Nie et al., 2023).*

In addition, we have added explanations for Equations 10 and 11. Lines 252-257 in the revised manuscript:

*Climate change contribution to BW and GW change is estimated by:*

$$CR_C = \frac{\left|X_2 - X_1\right|}{\left|X_2 - X_1\right| + \left|X_3 - X_2\right|} \times 100\% \tag{10}$$

*where $X_1$, $X_2$, and $X_3$ are the amount of water including BW or GWF and GWS, respectively for scenario S1, S2, and S3.*

*The contribution of LUCC to changes in BW and GW are estimated by* Equations 11.

$$CR_L = \frac{\left|X_3 - X_2\right|}{\left|X_3 - X_2\right| + \left|X_2 - X_1\right|} \times 100\% \tag{11}$$

*2. Language and Readability: The overall readability of the English text needs improvement. For example, in Line 162, "Both stations had simulation streamflow ..." should be corrected to "Both stations had simulated streamflow ...". Similar issues with unclear English should be checked and corrected throughout the manuscript to ensure the text is polished and easily understandable.*

Response: Thanks for your good suggestion. We made great efforts to improve our writing. We asked an English-specialist colleague to proof-read our final manuscript

to eliminate language problem as many as possible. All the changes were given in the marked version.

*3. Abbreviation Clarity: The use of abbreviations in the paper often feels unnatural and can be confusing. Typically, abbreviations are created using the initial letters of the terms they represent. Abbreviations such as LUCC (Land Use Change) and BWR (BW withdrawals) do not follow this convention and may lead to confusion. Clear and consistent use of abbreviations is necessary.*

Response: We have normalized the abbreviations.

Table 1 List of abbreviations

| Abbreviation | Full name | Abbreviation | Full name |
|---|---|---|---|
| *BW* | Blue water | *GW* | Green water |
| *GWF* | Green water flow | *GWS* | Green water storage |
| *BWSC* | Blue water scarcity | *GWSC* | Green water scarcity |
| *EFR* | Environmental flow requirements | *BWW* | Blue water withdrawals |
| *BWA* | Blue water availability | *GWFO* | Green water footprint |
| *GWA* | Green water availability | *P* | Precipitation |
| *T* | Temperature | *PET* | Potential evapotranspiration |
| *ET* | Evapotranspiration | LUCC | Land use and land cover change |

*4. Contradictions in Precipitation Trends: It is evident from Figure 3a that there are no significant increases or decreases in precipitation trends. However, the Discussion*

*and main text cite precipitation trends as reasons for certain results, which appears*

*contradictory. It would be more appropriate to present a figure showing statistically*

*significant trends in precipitation and base the discussion on those results. Additionally,*

*Table 3 is unclear and requires revision for better comprehension.*

Response: Although from the point of view of the stations, the trend of changes in precipitation in the Dongjiang River basin is not statistically significant (Figure 3a). The average precipitation of Dongjiang River basin can be obtained from the precipitation of these station using the Voronoi method. The average precipitation in Dongjiang River basin decreased at a rate of 0.51 mm year$^{-1}$ ($p>0.05$) (Figure S3). Since the change in average precipitation is not statistically significant, we have revised the discussion on precipitation change. We have added descriptions of temporal changes in mean precipitation, temperature, and potential evapotranspiration in the basin (lines 335-340 in the revised manuscript), and removed the discussion of the relationship between precipitation and total water resources (lines 498-500 in the revised manuscript).

*The mean precipitation, temperature, and potential evapotranspiration of DRB*

*can be obtained from the precipitation, temperature, and potential evapotranspiration*

*of stations using the Tyson polygon method. The inter-annual variation of annual*

*precipitation in DRB showed an insignificant decreasing trend (-0.51mm a$^{-1}$). The*

*annual mean temperature showed a significant increasing trend (0.024℃ a$^{-1}$). The*

*annual potential evapotranspiration showed a significant decreasing trend (-0.38mm*

*a$^{-1}$) (Figure S3).*

[Figure]

Figure S3. Interannual variation of (a) precipitation, (b) temperature, and (c) potential evapotranspiration in the Dongjiang River basin from 1970 to 2017.

*5. Additional References: I recommend adding the following papers to the citation in Line 24 to enhance the literature review and context:*

- *S. Berezovskaya et al. (2004), DOI: 10.1029/2004gl021277*

- *Suzuki et al. (2021), DOI: 10.3390/rs13214389*

Response: We have added the references in the literature review. Lines 24-29 in the revised manuscript:

*Land use and land cover change (LUCC) and climate variability may alter hydrological processes in watersheds* (Berezovskaya et al., 2004; Chagas et al., 2022; Konapala et al., 2020; Tan et al., 2022)*, which successively affect variations of regional water resources (Hoek van Dijke et al., 2022; Pokhrel et al., 2021; Stocker et al., 2023; Suzuki et al., 2021), potentially leading to ecosystem degradation and severe water shortage crises (Aghakhani Afshar et al., 2018; Zuo et al., 2015).*

*By addressing these concerns, the manuscript's quality and clarity will be significantly improved, making it more accessible and informative to the readers.*

**References**

Aghakhani Afshar, A., Hassanzadeh, Y., Pourreza-Bilondi, M., Ahmadi, A., 2018. Analyzing long-term spatial variability of blue and green water footprints in a semi-arid mountainous basin with MIROC-ESM model (case study: Kashafrood River Basin, Iran). Theoretical and Applied Climatology 134, 885–899. https://doi.org/10.1007/s00704-017-2309-0

Berezovskaya, S., Yang, D., Kane, D.L., 2004. Compatibility analysis of precipitation and runoff trends over the large Siberian watersheds. Geophysical Research Letters 31. https://doi.org/10.1029/2004GL021277

Chagas, V.B.P., Chaffe, P.L.B., Blöschl, G., 2022. Climate and land management accelerate the Brazilian water cycle. Nat Commun 13, 5136. https://doi.org/10.1038/s41467-022-32580-x

Ding, B., Zhang, J., Zheng, P., Li, Z., Wang, Y., Jia, G., Yu, X., 2024. Water security assessment for effective water resource management based on multi-temporal blue and green water footprints. Journal of Hydrology 632, 130761. https://doi.org/10.1016/j.jhydrol.2024.130761

Hoek van Dijke, A.J., Herold, M., Mallick, K., Benedict, I., Machwitz, M., Schlerf, M., Pranindita, A., Theeuwen, J.J.E., Bastin, J.-F., Teuling, A.J., 2022. Shifts in regional water availability due to global tree restoration. Nature Geoscience 15, 363–368. https://doi.org/10.1038/s41561-022-00935-0

Hordofa, A.T., Leta, O.T., Alamirew, T., Chukalla, A.D., 2023. Climate Change Impacts on Blue and Green Water of Meki River Sub-Basin. Water Resour Manage 37, 2835–2851. https://doi.org/10.1007/s11269-023-03490-4

Konapala, G., Mishra, A.K., Wada, Y., Mann, M.E., 2020. Climate change will affect global water availability through compounding changes in seasonal precipitation and evaporation. Nat Commun 11, 3044. https://doi.org/10.1038/s41467-020-16757-w

Nie, N., Li, T., Miao, Y., Zhang, W., Gao, H., He, H., Zhao, D., Liu, M., 2023. Asymmetry of blue and green water changes in the Yangtze river basin, China, examined by multi-water-variable calibrated SWAT model. Journal of Hydrology 625, 130099. https://doi.org/10.1016/j.jhydrol.2023.130099

Pokhrel, Y., Felfelani, F., Satoh, Y., Boulange, J., Burek, P., Gädeke, A., Gerten, D., Gosling, S.N., Grillakis, M., Gudmundsson, L., Hanasaki, N., Kim, H., Koutroulis, A., Liu, J., Papadimitriou, L., Schewe, J., Müller Schmied, H., Stacke, T., Telteu, C.-E., Thiery, W., Veldkamp, T., Zhao, F., Wada, Y., 2021. Global terrestrial water storage and drought severity under climate change. Nat. Clim. Chang. 11, 226–233. https://doi.org/10.1038/s41558-020-00972-w

Stocker, B.D., Tumber-Dávila, S.J., Konings, A.G., Anderson, M.C., Hain, C., Jackson, R.B., 2023. Global patterns of water storage in the rooting zones of vegetation. Nat. Geosci. 1–7. https://doi.org/10.1038/s41561-023-01125-2

Suzuki, K., Park, H., Makarieva, O., Kanamori, H., Hori, M., Matsuo, K., Matsumura, S., Nesterova, N., Hiyama, T., 2021. Effect of Permafrost Thawing on Discharge of the Kolyma River, Northeastern Siberia. Remote Sensing 13, 4389.

https://doi.org/10.3390/rs13214389

Tan, X., Wu, X., Huang, Z., Deng, S., Hu, M., Yew Gan, T., 2022. Detection and attribution of the decreasing precipitation and extreme drought 2020 in southeastern China. Journal of Hydrology 610, 127996. https://doi.org/10.1016/j.jhydrol.2022.127996

Veettil, A.V., Mishra, A.K., 2018. Potential influence of climate and anthropogenic variables on water security using blue and green water scarcity, Falkenmark index, and freshwater provision indicator. Journal of Environmental Management 228, 346–362. https://doi.org/10.1016/j.jenvman.2018.09.012

Zuo, D., Xu, Z., Peng, D., Song, J., Cheng, L., Wei, S., Abbaspour, K.C., Yang, H., 2015. Simulating spatiotemporal variability of blue and green water resources availability with uncertainty analysis. Hydrological Processes 29, 1942–1955.

---

## Author Comment (AC2)

*General Comments*

*The manuscript "Combined Impacts of Climate Change and Human Activities on Blue and Green Water Resources in the High-Intensity Development Watershed" presents an intriguing analysis of the variations in blue water (BW) and green water (GW) resources in the study area.*

We wish to sincerely thank the reviewers for their extensive and thoughtful comments on our manuscript which we have addressed in the revised manuscript as discussed below. Throughout, reviewer comments are in *blue* font and *italic* type, and **our response** in **black** font.

*Major Comments:*

*Readability (Grammar): The overall readability of the English text needs improvement. There are several grammatical issues and problems that complicate the readability of the text. I suggest a thorough review and editing of the text to enhance its clarity and fluency before it can be considered for publication.*

Response: Thanks for your good suggestion. We have made great efforts to improve our writing. We asked an English-specialist colleague to proof-read our final manuscript to eliminate language problem as many as possible. All the changes were given in the marked version.

*Literature Review: The literature review lacks some recent works that have also analyzed the effects of climate change and landscape change on the water cycle. Specifically, a refined search for studies using the SWAT model would reveal many works that should be mentioned in the introduction to provide a more comprehensive background.*

Response: We have added some recent references in the literature review. Lines 84-97 in the revised manuscript:

*Water resources management is the primary issue to be addressed for water security. Hydrological models are important tools to meet various needs in water resource management. Hydrological model simulation is an effective method to*

*evaluate changes in blue and green water resources. As a widely used semi-distributed parametric hydrological model, the SWAT model, which typically subdivides watershed into smaller subbasins, is increasingly used in water resources management at the watershed scale. Based on the SWAT model, researchers simulated the spatiotemporal changes in blue and green water resources in Iran (Jeyrani et al., 2021), the Yangtze River basin (Nie et al., 2023), the Poyang Lake basin (Liu et al., 2023), India (Sharma et al., 2023). Some studies have also used model simulations to analyze the effects of climate change and human activities on water resource changes in China (Liu et al., 2022), Meki River basin (Hordofa et al., 2023), and Ningxia (Wu et al., 2021), etc. However, most of the hydrological models used in the study were calibrated and validated using only observed streamflow data without checking the accuracy of other simulated water variables, which can lead to uncertainties in modeling soil moisture and evapotranspiration (Nie et al., 2023).*

*Presentation of Results: The results of the calibration and validation are currently presented in the methods section. These should be moved to the results section for better coherence and logical flow of the manuscript.*

Response: We have moved the results of the calibration and validation to the results section.

*Scenario Definition: The definition of the three scenarios is still confusing. Please clarify how each scenario was considered and defined to ensure readers can easily understand the distinctions and implications of each scenario.*

Response: We have added the definitions of the three scenarios. To distinguish the single and combined effects of land use change and climate change on the water resources of DRB, three scenarios listed below were established in this study. The land use map was fixed when simulating the influences of climate change on blue and green

water (S2-S1), while climate conditions was fixed when simulating the influences of LUCC on blue and green water (S3-S2). The climate conditions and the land use were altered when assessing the joint influences of climate change and LUCC on blue and green water (S3-S1). Lines 194-201 in the revised manuscript:

*Three scenarios were constructed to assess the impacts of climate change and LUCC on BW and GW by changing climate conditions (land use) while holding land use (climate conditions) for the three scenarios simulation each (Table 2). The land use map was fixed when simulating the influences of climate change on blue and green water (S2-S1), while climate conditions was fixed when simulating the influences of LUCC on blue and green water (S3-S2). The climate conditions and the land use were altered when assessing the joint influences of climate change and LUCC on blue and green water (S3-S1).*

Table 2 Scenario settings for the simulation of effects of climate change and LUCC on blue and green water

| Scenarios | Land use | Climate period | Combined effects | Land use change effects | Climate change effects |
|---|---|---|---|---|---|
| S1 | 1980 | 1970-1993 | | | |
| S2 | 1980 | 1994-2017 | | | S2-S1 |
| S3 | 2015 | 1994-2017 | S3-S1 | S3-S2 | |

*By addressing these concerns, the manuscript can be further evaluated and considered for publication.*

**References**

Hordofa, A.T., Leta, O.T., Alamirew, T., Chukalla, A.D., 2023. Climate Change Impacts on Blue and Green Water of Meki River Sub-Basin. Water Resour Manage 37, 2835–2851. https://doi.org/10.1007/s11269-023-03490-4

Jeyrani, F., Morid, S., Srinivasan, R., 2021. Assessing basin blue–green available water components under different management and climate scenarios using SWAT. Agricultural Water Management 256, 107074. https://doi.org/10.1016/j.agwat.2021.107074

Liu, M., Wang, D., Chen, X., Chen, Y., Gao, L., Deng, H., 2022. Impacts of climate variability and land use on the blue and green water resources in a subtropical basin of China. Sci Rep 12,

20993. https://doi.org/10.1038/s41598-022-21880-3

Liu, M., Zhang, P., Cai, Y., Chu, J., Li, Y., Wang, X., Li, C., Liu, Q., 2023. Spatial-temporal heterogeneity analysis of blue and green water resources for Poyang Lake basin, China. Journal of Hydrology 617, 128983. https://doi.org/10.1016/j.jhydrol.2022.128983

Nie, N., Li, T., Miao, Y., Zhang, W., Gao, H., He, H., Zhao, D., Liu, M., 2023. Asymmetry of blue and green water changes in the Yangtze river basin, China, examined by multi-water-variable calibrated SWAT model. Journal of Hydrology 625, 130099. https://doi.org/10.1016/j.jhydrol.2023.130099

Sharma, A., Patel, P.L., Sharma, P.J., 2023. Blue and green water accounting for climate change adaptation in a water scarce river basin. Journal of Cleaner Production 426, 139206. https://doi.org/10.1016/j.jclepro.2023.139206

Wu, J., Deng, G., Zhou, D., Zhu, X., Ma, J., Cen, G., Jin, Y., Zhang, J., 2021. Effects of climate change and land-use changes on spatiotemporal distributions of blue water and green water in Ningxia, Northwest China. J. Arid Land 13, 674–687. https://doi.org/10.1007/s40333-021-0074-5

---

## Author Response (AR2)

*Dear Authors,*

*Thank you for submitting the revised version of your manuscript. We appreciate the effort you have put into addressing comments and suggestions. After reviewing the revised manuscript, we are pleased to note that it has improved significantly, and your revisions have enhanced the clarity of the work.*

*In addition, feedback from an additional reviewer is now available, requiring minor revisions. Please review the comments carefully and incorporate the necessary changes to strengthen your manuscript further.*

*We look forward to receiving your response and the revised manuscript.*

*Best regards, Elham Freund*

Dear Elham Freund,

We wish to thank you for handling the review of our manuscript submitted to HESS for possible publication. We wish to sincerely thank the reviewers for their extensive and thoughtful comments on our manuscript which we have addressed in the revised manuscript as discussed below. Throughout, *reviewer comments* are in *blue* font and *italic* type, and **our response** in **black** font. OM and RM stand for original and revised manuscript, respectively.

There have been textual changes throughout the manuscript, mostly in Method, Results and Discussion. All the changes were given in the marked version.

Thanks a lot for your consideration.

Thank you and with regards.

Sincerely,

Xuezhi Tan

*The topic selection of this paper holds significant practical significance, focusing on the blue and green water resources in intensively developed watersheds, which is crucial for understanding and responding to challenges in global water resource management. The study employs methods such as hydrological model simulation, statistical analysis, and cluster analysis to examine the combined impacts of climate change and human activities on blue and green water resources at both the watershed and sub-basin scales. Using three water security indices, this study comprehensively assesses the water resource condition of the Dongjiang River Basin, explains the dynamic changes of blue and green water resources in intensively developed watersheds, and provides a new theoretical perspective for water resource management. The innovations include the establishment of the SWAT model through multi-water flux calibration and verification, and the exploration of the combined impacts of climate change and land use change on water resources from the perspectives of blue and green water. The manuscript is well organized, and the methods are robust. Overall, I would recommend a minor revision for this manuscript. The detailed comments are given below.*

*Major comments:*

*1. Methods: More details on the simulation process should be added, such as potential evapotranspiration calculation methods, the surface runoff process, etc.*

Response: We have added the details of the SWAT model. Lines 149-153 in the RM:

*The SCS curve number method was used for flow partitioning according to land use, soil type and antecedent soil moisture. The Penman-Monteith method was used to calculate potential evapotranspiration, which comprehensively considered various climatic factors such as solar radiation, air temperature, wind speed and relative humidity (Arnold et al., 1998; Neitsch et al., 2002).*

*2. Line 210: Although the blue and green water scarcity index have been defined here, how can we assess the degree of blue and green water scarcity in the basin according to these indices? A description of the blue and green water shortage classification should be added.*

Response: We have added the description of the blue and green water scarcity

classification and thresholds. Lines 215-217 in the RM:

*Based on the blue water scarcity and green water scarcity, water scarcity of a region is categorized as: mild scarcity, moderate scarcity, severe scarcity and extreme scarcity, with thresholds set at 100%, 150% and 200%, respectively.*

*3. Results: The model was calibrated and verified by using the reconstructed natural streamflow, but only the method of streamflow reconstruction was introduced. Comparative analysis of observed and natural streamflow should be added.*

Response: We have added the comparative analysis of observed and natural streamflow. Lines 273-285 in the RM:

*3.1.1 Streamflow reconstructed*

*The difference between the monthly average observed streamflow and the monthly average natural streamflow is small (Figure 2). The monthly average measured streamflow and natural streamflow at the Heyuan station is 492.1 $m^3$ $s^{-1}$ and 507.9 $m^3$ $s^{-1}$, respectively, while the monthly average measured streamflow and natural streamflow at the Boluo station is 768.4 $m^3$ $s^{-1}$ and 796.7 $m^3$ $s^{-1}$, respectively. The difference between the measured streamflow and the natural streamflow mainly occurs in November, December, January, and February (where the measured streamflow is greater than the natural streamflow) and May, June, and July (where the measured streamflow is less than the natural streamflow) (Fig. 2a and Fig. 2c).*

[Figure]

Figure 2. Observed streamflow and natural streamflow processes at the Heyuan and Boluo stations from 1970 to 2000. (a) Annual distribution of streamflow at the Heyuan station, (b) streamflow process at the Heyuan station, (c) annual distribution of streamflow at the Boluo station, (d) streamflow process at the Boluo station

*4. Lines 357-366: The Dongjiang River Basin is located in the subtropical monsoon climate zone, and the distribution of water and heat is uneven throughout the year. It might be better to add seasonal variations in blue and green water and its scarcity.*

Response: We have added the seasonal variations of blue and green water scarcity in the supplementary materials.

*1 Seasonality variation of blue and green water scarcity*

*The time of occurrence of blue and green water scarcity in the basin during the year is different, with the peak of blue water scarcity occurring from October to March, while green water scarcity mainly occurs from May to September (Figure S5). The climate of the Dongjiang River Basin belongs to the subtropical monsoon climate, and precipitation is mostly concentrated in the flood season (April to September), resulting in larger river streamflow from April to September and larger blue water resources available in the basin; The available blue water resource is low in the dry season (October to March), so moderate, severe, and extreme blue water scarcity occurs in the downstream sub-basins with a large population during the dry season. The population in the upstream sub-basins is smaller, so the risk of blue water scarcity is smaller. It is worth noting that this study only distributes the annual blue water demand evenly to each month and does not consider the intra-year change in blue water demand, which may cause certain errors in the results. Green water demand tends to be smaller from October to April, while vegetation growth is strong from May to September, and therefore evapotranspiration from the watershed is larger, based on the results in the previous section green water storage (soil moisture) fluctuates within the year much less than evapotranspiration (green water streamflow), resulting in moderate green water scarcity in May to September in the four sub-watersheds of the middle reaches of the watershed.*

[Figure]

Figure S5 Intra-annual variation of blue and green water scarcity in each sub-basin of Dongjiang River basin.

*5. Lines 400-418: It is more interesting to compare the differences in blue and green water shortages across sub-basins.*

Response: We have added the blue and green water scarcity in each sub-basin. Lines 438-442 in the RM:

*Figure S4 shows the annual variation of blue water scarcity and green water scarcity in the basin. Except for some sub-basins, the blue and green water scarcity in most sub-basins is less than 50%. The degree of green water scarcity is higher than that of blue water scarcity in most of the sub-basins. Only the sub-basin 63 in downstream experienced a severe blue water scarcity.*

[Figure]

Figure S4 Annual average blue water scarcity and green water scarcity in each sub-basin of the Dongjiang River basin.

*6. Discussion: When quantifying the scarcity of blue water, a coefficient of 0.8 was used to represent the proportion of environmental flow in blue water resources. Should the coefficient be adjusted in different wet and dry basins? Whether the use of varying coefficient ratios will affect the results.*

Response: We have added the discussion of the effect of threshold for environmental flow on the results. Lines 561-569 in the RM:

*(3) Both the calculations of BWSC and the FLK index include environmental flows. This study represented the proportion of environmental flow in streamflow as 80%. Some studies have suggested that assuming environmental flow to be 80% of the total water resources in a basin may overestimate water scarcity (Liu et al., 2017; Richter et al., 2012). Therefore, we varied the proportion of environmental flow and assessed the degree of BWSC using 60% and 70% proportions. Results show that only the 63rd sub-basin changed from severe BWSC to moderate to severe BWSC, while other sub-basins remained with low BWSC. Therefore, the threshold for environmental flow has a minor impact on this paper.*

*Minor comments:*

*1. Line 16: change "have" to "has".*

Corrected.

*2. Line 87: "India" should be "and India".*

Corrected.

*3. Line 116: change "of" to "in".*

Corrected.

*4. Line 210: Please check the period here.*

Corrected.

*5. Line 356 and 398: delete "during".*

Corrected.

*6. Line 470: change "was" to "were".*

Corrected.

*7. Line 551: "restriction is" should be "restrictions are".*

Corrected.

**References**

Arnold, J. G., Srinivasan, R., Muttiah, R. S., and Williams, J. R.: Large Area Hydrologic Modeling and Assessment Part I: Model Development1, JAWRA Journal of the American Water Resources Association, 34, 73–89, https://doi.org/10.1111/j.1752-1688.1998.tb05961.x, 1998.

Liu, J., Yang, H., Gosling, S. N., Kummu, M., Flörke, M., Pfister, S., Hanasaki, N., Wada, Y., Zhang, X., Zheng, C., Alcamo, J., and Oki, T.: Water scarcity assessments in the past, present, and future, Earth's Future, 5, 545–559, https://doi.org/10.1002/2016EF000518, 2017.

Neitsch, S., Arnold, J., Kiniry, J., Williams, J., and King, K.: Soil and water assessment tool (SWAT): theoretical documentation, version 2000, Texas Water Resources Institute, College Station, Texas, TWRI Report TR-191, 2002.

Richter, B. D., Davis, M. M., Apse, C., and Konrad, C.: A Presumptive Standard for Environmental Flow Protection, River Research and Applications, 28, 1312–1321, https://doi.org/10.1002/rra.1511, 2012.